# Marketing and Distribution System Foster Misuse of Antibiotics in the Community: Insights from Drugs Wholesalers in India

**DOI:** 10.3390/antibiotics11010095

**Published:** 2022-01-13

**Authors:** Anita Kotwani, Arti Bhanot, Girdhari Lal Singal, Sumanth Gandra

**Affiliations:** 1Department of Pharmacology, Vallabhbhai Patel Chest Institute, University of Delhi, Delhi 110007, Uttar Pradesh, India; 2Independent Consultant, New Delhi 110070, Uttar Pradesh, India; arti.bhanot@gmail.com; 3State Drugs Controller (ex) Food and Drugs Administration, Panchkula 134109, Haryana, India; girdharilalsingal@yahoo.com; 4Department of Internal Medicine, Division of Infectious Diseases, Washington University School of Medicine, St. Louis, MO 63110, USA; gandras@wustl.edu

**Keywords:** antibiotics, antimicrobial resistance, drug wholesaler, informal health care provider, marketing, pharmaceutical industry, wholesaler, medical representatives

## Abstract

Antibiotic misuse is one of the major drivers of antimicrobial resistance (AMR). In India, evidence of antibiotic misuse comes largely from retailers as well as formal and informal healthcare providers (IHCPs). This paper presents the practices and perspectives of drug wholesalers, a critical link between manufacturers and last-mile dispensers. Four experienced wholesalers and an ex-State Drug Controller (ex-SDC) were interviewed in depth, using semi-structured guides in the National Capital Region of Delhi, India, between November 2020 and January 2021. Four main findings were that wholesalers (i) have limited knowledge about wholesale licensing and practice regulations, as well as a limited understanding of AMR; (ii) directly supply and sell antibiotics to IHCPs; (iii) facilitate medical representatives (MRs) of pharmaceutical companies and manufacturers in their strategies to promote antibiotics use in the community; and (iv) blame other stakeholders for unlawful sale and overuse of antibiotics. Some of the potential solutions aimed at wholesalers include having a minimum education qualification for licensing and mandatory Good Distribution Practices certification programs. Decoupling incentives by pharmaceutical companies from sales targets to improve ethical sales practices for MRs and optimize antibiotic use by IHCPs could alleviate wholesalers’ indirect actions in promoting antibiotic misuse.

## 1. Introduction

Antibiotic misuse is common in low- and middle-income countries (LMICs) [1,2,3,4], and this practice is one of the major drivers of antimicrobial resistance (AMR). Inequalities in healthcare, wide economic disparity, corruption resulting in poor return on taxation, poor hygiene and sanitation systems, easy availability of over-the-counter (OTC) antibiotics, self-medication in the community, preference for broad-spectrum antibiotics, lack of human resources for health, widespread use of antibiotics in animals, and poor regulation and implementation are dominant reasons for antibiotic misuse in LMICs [5,6]. Antibiotic consumption in India increased from 3.2 billion defined daily doses (DDDs) in 2000 to 6.5 billion DDDs in 2015, and remains the highest in the world [6]. In India, there is an increasing trend in the overuse of antibiotics—especially broad-spectrum and newer antibiotics [7]. Concern has been expressed about irrational prescription practices by physicians and informal healthcare providers (IHCPs) [8,9,10,11]. Drug promotion and marketing tactics by pharmaceutical companies also play an important role in driving the OTC sale of antibiotics and indiscriminate prescription practices by physicians and IHCPs [12,13].

In India, the manufacture, sale, stock, or distribution of medicines is regulated under the Drugs and Cosmetics Act (DCA), 1940, and the Drugs and Cosmetics Rules (DCR), 1945 [14]. As per the DCA and DCR, all antibiotics are prescription drugs and come under category Schedule H (Drugs and Cosmetic Rules, 1940). An amendment to Schedule H was made in 2014 to include the second- and third-generation antibiotics into a new Schedule H1 category, which requires retail pharmacists to maintain a separate register for the sale of these antibiotics (Drugs and Cosmetic Rules, 1940). Qualitative studies in India provide evidence of the inappropriate use of antibiotics by consumers [15], OTC sale by pharmacists [16,17], doctors’ perceptions about the demand of antibiotics by patients [18], and inherent misconceptions and marketing influences on antibiotic over prescription by IHCPs [19]. However, in this supply and use chain of antibiotics, insights from wholesalers in the trade of distributing antibiotics to many stakeholders (e.g., retailers, hospitals, and dispensing doctors (including IHCPs)) is not known. Wholesalers are licensed traders, approved under the DCR for the sale and distribution of medicines to retailers, hospitals, and healthcare providers. In India, central drug laws (DCR) are applicable to all states and union territories (UTs) uniformly. Therefore, the legal provisions relating to wholesale licensing and practices are similar in all states/UTs. Stockists and sub-stockists identified by pharmaceutical companies to stock and sell their drugs are also considered wholesalers. The main objective of this qualitative study was to obtain insights from experienced wholesalers in the pharmaceutical trade on the distribution practices of antibiotics and to explore their role in the misuse of antibiotics in the community, as well as their understanding of AMR.

## 2. Methodology

### 2.1. Study Design and Setting

This qualitative study was conducted in the National Capital Region (NCR) of Delhi, India, which includes National Capital territory (NCT) and surrounding cities of neighbouring states [20]. In-depth interviews were conducted with four wholesalers who are in the business of pharmaceuticals for more than 20 years to explore the practices regarding antibiotics’ sale in the community and their awareness of AMR. The two satellite towns adjacent to NCT, Delhi included in this study had rural and urban areas and citizens staying or working in these cities or in NCT, Delhi. The study was conducted between 18 November 2020 and 31 January 2021. Considering the sensitive nature of information shared by the respondents, it was critical for the researchers to have a trusting relationship to gain unrestricted responses. Thus, this study focused on a small group of respondents.

### 2.2. Study Participants

One of the respondents (W1) was from a satellite town of NCR, Delhi, managing an ancestral wholesale business of medicines for the past 30 years. W1 possess distributorship of the medicines of 20 to 25 companies and supply medicines to about 1200 retailers. A second respondent from another satellite town of NCR, Delhi (W2), has been in this business for the past 31 years. W2 also has distributorship of 20–21 companies and supply to 1300–1400 retailers. W3 is in retail pharmacy, wholesale, and has a small company promoting generics including antibiotics manufactured by a third party. W3 supply medicines to 200–250 retail pharmacies and also directly supply medicines including antibiotics to doctors who dispense their prescribed medicines. Finally, W4 is involved in the trade of medicines through retail, wholesale, and marketing for 35 years. W4 is involved in distribution and marketing for a multinational company and several Indian companies, including start-ups. He has a well-defined network of stockists in various places, especially in NCR, Delhi. W4 is also in the business of data collection on the sale and distribution of medicines from wholesalers and retailers from the entire NCR, Delhi. W1 is a qualified pharmacist, and the other three (W2, W3, and W4) have no formal training in pharmacy. All four distributed and dealt with only human medicines.

AK first reached out to W4 for the in-depth interviews due to earlier research association and respondent’s willingness to participate without reservations. Respondent W4 connected the author to another respondent W3 and through W3 two other respondents were identified and interviewed. The author (AK) reached out to three more wholesalers in NCR, Delhi, but they declined, indicating that they only distribute antibiotics or medicines to licensed retailers and that there is no misuse of antibiotics from their end.

In addition, a retired State Drug Controller (ex-SDC) (GLS, co-author) of one of the neighbouring states of Delhi was consulted for insights on medicine wholesale regulations and practices. A drug controller is appointed by the Government in each state/UT to monitor and ensure compliance to DCA/DCR.

### 2.3. Data Collection

All interviews were conducted online due to COVID-19 pandemic restrictions through Google Meet (Google LLC, Mountain View, CA, USA) with a prior appointment with the respondents made by AK. A brief introduction about the study was given and any queries were answered. Data were collected through in-depth interviews, using a semi-structured interview guide developed based on the scientific literature review and authors’ previous studies with consumers and retail pharmacists on the use and knowledge of antibiotics and AMR.

The interview guide was used to explore topics related to drug licensing and regulations for wholesale business, knowledge of antibiotic use, exploring insight for promoting the use of antibiotics in the community, business strategies to promote the sale of antibiotics and marketing pressure from pharmaceutical industry representatives. The interview guide was continuously modified based on emerging themes from the interviews. The interviews lasted from 45 to 80 min. The interviews were conducted until saturation was reached.

Informed consent was taken before conducting the interview. Hindi and English mix was the language of discourse for interviews by AK and notes were taken by both AK and a social scientist (AS) who was working as research fellow on another project with AK. In concurrence with ethics considerations, informed verbal consent was obtained from the participants before the interviews, and interviews were also audio-recorded with the participants’ consent. The audio-recorded interviews were further transcribed and translated for analysis. The anonymity of the respondents was maintained for analysis and reporting.

### 2.4. Data Analysis

Interview recordings of the study were transcribed verbatim and translated into English in Microsoft Word (Redmond, WA, USA) for the analysis. The data were coded in Microsoft Word, and Microsoft Excel (Redmond, WA, USA) sheets were used for thematic analysis. The method described by Braun and Clarke (2006) [21] was used, which involves becoming familiarized with the data through an iterative process of reading the dataset transcripts, generating initial codes, arranging codes into larger categories, and drawing connections between codes and categories until the generation of a saturated thematic map of the analysis. The initial data coding was conducted by the social scientist (AS), and the themes derived were discussed with AK. They were revised and refined through regular meetings in the first phase, and revisions were made wherever necessary to mitigate interpretative bias. All major themes and sub-themes were agreed with discussion and consensus of the entire team (AK, AS, SG, and AB). Finally, a round of discussion was held among the four authors, AK, AB, GLS, and SG, to finalize the themes and sub-themes. A combination of both deductive and inductive approaches was used in coding the data.

## 3. Results

A total of four themes and sub-themes emerged from the interviews (Table 1). Under each theme, findings from interviews with the four wholesalers and the ex-SDC are presented.

### 3.1. Rules and Regulations Governing Wholesale Licensing and Practice

This theme presents rules, regulations, and guidelines for wholesalers as elucidated by the interviewees.

#### 3.1.1. Procurement of a Wholesale License

All wholesalers mentioned that a wholesaler or a stockist requires a license for the stock/supply/distribution of medicines including antibiotics. Two wholesalers (W3 and W4) mentioned a minimum experience of 2–3 years in pharmaceutical trade practice is mandatory for license. One wholesaler (W4) specified that candidates with education up to 12th grade need 2 years of experience; graduates require one year of experience in pharmaceutical trade, and no experience is necessary if the applicant is a pharmacist (diploma in pharmacy or pharmacy graduate). One of the respondents (W4) mentioned the dissolution of the renewal procedure as per the amended DCR; however, the respondent added that a retention fee needs to be paid every five years for continuity of the license. This was reinforced by another respondent (W3), who mentioned a license continuity fee of about USD 40 (INR 3000) needs to be paid every five years.

Respondents W1 and W2 had differing views on wholesale licensing compared to W3 and W4. W1 mentioned a diploma in pharmacy as a mandatory requirement for wholesale licensing. Respondent W2 mentioned that a diploma in pharmacy is preferred by wholesalers to avoid administrative bottlenecks involved in getting an experience certificate as per the recent requirements.


*“Earlier it was like anyone with a simple experience certificate used to get the license for wholesale trade but now they have these requirements, how you got into sales, how you got into distribution, they may ask for payslip from employer etc. Sometimes people are not able to produce these documents. So now people think that let’s have a diploma in pharmacy so that we can start immediately.”*
(W2)

The ex-SDC mentioned that the DCA 1940 and DCR 1945 have provisions for two types of licenses for the sale of drugs. One is the wholesale drugs license wherein drugs are sold/distributed/supplied to retailers/hospitals/doctors for further sale or supply to patients. Carry and forwarding (C & F) agents, distributors, marketing firms, and wholesalers are some examples of wholesale drugs license holders. The two main requirements for a wholesale license are that licensees: (1) have storage facilities (premises) including cold chain management and (2) are a “competent person” who can be an employee, owner, partner, director, or trustee. A competent person is considered one with a 10th grade education with a minimum of four years’ experience in drug sales, one with any university degree and one year experience in drug sales, or a registered pharmacist. Hence, having a diploma in pharmacy is not mandatory. The latter is mandatory to get a retail drug license, which is the second type of license issued under the DCR. The ex-SDC validated the deletion of provisions for license renewal for both retailers and wholesalers subject to the regular and timely deposition of retention fees.

#### 3.1.2. Guidelines for Wholesaling Practice

W1 and W2 mentioned that the guidelines for the sale of medicines by pharmacists and wholesalers were the same. These respondents mentioned that while dispensing Schedule H1 drugs, the wholesaler or retailer has to maintain a separate register with details of the sale of such drugs. However, wholesalers were largely unaware of this requirement, and very few had the inclination to follow it.

W3 and W4 mentioned receiving guidelines forbidding the sale/supply of drugs to IHCPs. W1 also confirmed this but mentioned that it is not practiced.


*“People are not following it that is different but the guideline is there that we have to sell medicines only to a retailer or to MBBS doctors and we should not give to unqualified IHCPs.”*
(W1)

The ex-SDC clarified that recording details of the sale of H1 drugs in a separate register is mandatory for retailers but not for wholesalers. He affirmed that there are frequent violations of this rule, with most retailers not maintaining registers of H1 drug prescriptions. He also validated the wholesalers’ information on the rampant sale of drugs to IHCPs.

#### 3.1.3. Role of Drug Inspectors and Inspections

Respondent W3 mentioned that inspections were held annually, and that inspectors mainly checked sale and purchase records. Respondents hinted at a lenient behaviour towards wholesale practice.


*“Wholesalers have to show their sale and purchase only and if that is getting matched then there is no problem.”*
(W3)

Responses of W2 were contrasting, stating that inspections include full review of the wholesale premises. Samples are collected, and standing orders for holding sale are given for certain medicines until a clearance is given by the regulator. W2 mentioned that drug inspectors had a lot of power and could even cancel the license.


*“They (drug inspectors) come and check like they go to a retailer and there also billing is there so they check everything that everything is proper or not. Whether supply is proper or not, the cold chain is maintained or not so they check all these things.”*
(W2)

W1 mentioned that the criteria for drug inspections vary with places or cities. Some may be coerced to overlook irregularities but if there are gross irregularities, inspectors follow protocols.

The ex-SDC provided a detailed account of an inspection, which includes an inspection of storage facilities, cold chain in case of vaccines, and the appropriate storage of thermally labile drugs (e.g., antibiotics, vitamins), purchase records, stocking of prohibited medicines, government supply not for sale, physician samples, records of sale and purchase, and defects in sale invoices (whether sale was made with legal persons/retailers holding a drug license number and the registration numbers of doctors, whether written orders of doctors to whom drugs were supplied were made and maintained, or not). Inspectors can determine any unlawful sale or purchase from the scrutiny of records only. The latter makes it challenging to know the actual scenario, as records may be manipulated. Further, he mentioned an enormous gap in the required and actual number of inspectors.


*“As per Haathi committee (1975) there should be one inspector for 100 retail or wholesale units/25 drugs factories, but I think there is one for over 1000 shops. How can annual inspections happen?”*
(ex-SDC)

### 3.2. Antibiotic Use and Misuse in the Community

This theme presents the perception of wholesalers on antibiotics use in the community, and the role of doctors, IHCPs, consumers, and pharmacists in overuse. Wholesalers felt they were least accountable for the rampant use of antibiotics in the community, as they do not create demand, but only meet it.

#### 3.2.1. The Trend of Antibiotic Use in the Community

Two of the respondents (W1 and W2) commented on the increasing trend of antibiotic use in the community and changes in choice of antibiotics over years. Respondents stated that there has been a shift from tetracycline, chloramphenicol, and ciprofloxacin to antibiotics like amoxicillin-clavulanic acid, cefixime, and meropenem in both urban and rural areas.


*“In every 2–5 years trend keeps on changing like earlier ciprofloxacin was in use and today even meropenem also people are using very casually.”*
(W2)

#### 3.2.2. OTC Sale of Antibiotics among Pharmacists

Respondents W1 and W2 mentioned the OTC sale of antibiotics by pharmacists as a common practice. W1 explained that half of the pharmacists are not aware about the misuse of antibiotics, and that even if they are, they engage in the OTC sale of antibiotics or medicines from a business perspective. W4 mentioned that rules like maintenance of Schedule H1 registers by retailers are easy to manipulate and do not discourage the sale of antibiotics without prescriptions. Register entries are only made for invoice-linked sales, and the rest of the Schedule H1 drugs (mostly include antibiotics) are sold without a prescription. Furthermore, W4 emphasised that frequently misused antibiotics (e.g., azithromycin) are not even categorized under Schedule H1 drugs. Pharmacists are able to generate counterfeit invoice for without prescription drugs by simply putting the name of a nearby hospital, the name of a patient, and the quantity of the medicine dispensed, which is a safeguard during inspections.

Interestingly, the OTC sale of antibiotics is guided by doctors’ earlier prescriptions. Pharmacists tend to dispense antibiotics prescribed to one patient to another with similar complaints or to the same patient based on the precedence, without consulting the doctor.


*“Half of the chemists (retail pharmacy shops) are not aware about AMR and even if they know, they have to do business so they give antibiotics without prescription.”*
(W1)

The ex-SDC had similar views on pharmacists’ role in promoting misuse of antibiotics. In his experience, this malpractice was more common in smaller towns and villages rather than cities, as IHCPs are also more prevalent in smaller towns. Additionally, he quoted that many pharmacists desire profit from sale of generic drugs that are purchased at much lower costs.

W4 explained that there are two types of drugs in the Indian pharmaceutical market. Almost 99% in the true sense are generic medicines and have an assigned trade name, thus called branded-generics. Generally, the branded-generics are promoted in an ethical manner—that is, a prescription is generated and product promotion is done via the pharmaceutical company’s medical representatives; these branded-generics are popular trade names for that drug. However, the less-popular generics are sold directly to doctors or to retailers by wholesalers or manufacturers. Furthermore, another respondent (W3) elucidated that less popular generic medicines have a higher profit margin, and unsold stocks or those close to expiry are not taken back by the companies, while the popular branded-generics brands are taken back by the company if not sold at retail pharmacies.


*“For generic medicines they (retailers) have huge profit margins but these (generic medicines) cannot be returned.”*
(W3)

#### 3.2.3. Indiscriminate Use of Antibiotics by Doctors

Two respondents (W1 and W2) opined that doctors prescribe antibiotics for their patients to have quick relief from symptoms. W1 mentioned that broad-spectrum antibiotics (e.g., meropenem) are mostly prescribed for in-patients in hospital settings, but injectables are also being used indiscriminately. W4 believed that doctors were the main influencers in the rampant sale of antibiotics, as medical representatives (MRs) and wholesalers cannot create artificial demands. Further, W4 mentioned that doctors prescribe brands (popular trade names), not the molecule/pharmacological/generic names, and on one occasion he (W4) noted that two antibiotics of the same molecule (pharmacological agent) with two different trade names were prescribed.


*“I would say 90% of the doctors seldom remember the active ingredient in the brand (a trade name) they are writing more, this is more so with fixed-dose combination (FDC).”*
(W4)

#### 3.2.4. Misuse of Antibiotics by the IHCPs

The respondent mentioned that the practice of IHCPs in the community is an area of concern, especially in reference to antibiotic misuse. Two respondents (W1 and W2) mentioned that IHCPs more or less dispense antibiotics to every patient.


*“These IHCPs study the trend of prescription of a nearby hospital. If the hospital is prescribing ciprofloxacin of Cipla company then they study that particular salt (ciprofloxacin) and even identify some alternative company which provides generic, which have more profit margin and then prescribe or dispense that.”*
(W1)

W4 reiterated the rampant use of antibiotics by IHCPs. However, he mentioned that in low-income areas, medicines and treatments need to be given at a reasonable cost because of the affordability issue of such patients. Hence, IHCPs generally dispense medications for two days. The respondent explicitly emphasised that most IHCPs are unaware of the active ingredients of the medicines they are dispensing.

#### 3.2.5. Role of Wholesalers in the Misuse of Antibiotics

Two respondents (W1 and W2) stated that wholesalers are the indirect and least-important stakeholders responsible for the misuse of antibiotics in the community, as wholesalers procure medicines based on the demand from the retail pharmacists. However, W1 added that around 15–20% of the medicines sold by wholesalers are dispensed/supplied to IHCPs.


*“On a scale of 100 if I have to say then I think around 15–20% of it (antibiotics) we are selling to IHCPs and these products come under generic.”*
(W1)

Moreover, W4 mentioned that wholesalers are not very concerned about ethical use, as they are maintaining the stocks of medicines according to the need generated in the market. The respondent (W4) mentioned that wholesalers can only promote products due to discounts provided by the MRs, but they still supply the product based on the demand. Retailers purchase stock of low-cost medicines or discounted drugs (antibiotics) and then try to substitute for the popular trade-name medicines with every prescription. Moreover, the retailers promote the generics or antibiotics for which they have a greater profit margin when dispensing to patients who come directly to them for treatment or self-medication.

The ex-SDC also had similar views, that the problem of unauthorized medicine/antibiotic sale is localized among the few wholesalers dealing with IHCPs. Wholesalers trading with large pharmaceutical companies do not risk associations with IHCPs and generally do not pose a threat to sale among such providers.

#### 3.2.6. Consumers’ Inappropriate Antibiotic Consumption Practice

One respondent (W4) mentioned that consumers themselves use antibiotics indiscriminately. Patients tend to discontinue medication as soon as they feel better without completing the full course. As mentioned earlier, patients demand immediate relief from symptoms, which could influence doctors’ prescription, as dissatisfied patients may not return in the future and impact the doctors’ ‘businesses’. Furthermore, W4 strongly believed that self-medication practice drives antibiotic misuse in India. There is a tendency to rely on earlier prescriptions or those of other patients among family and friends, and to self-medicate in case of a similar ailment.


*“Everyone knows that if I am having this problem, I can take this medicine which he has received earlier or someone else had.”*
(W4)

### 3.3. Potential Role of MRs and Pharmaceutical Companies in Antibiotics Misuse

This theme represents how MRs play a key role in the trade of pharmaceutical products. This theme highlights the role of MRs in promoting medicines (including antibiotics) in the community via marketing strategies and incentives.

#### 3.3.1. MRs’ Target-Driven Approach to Marketing

One of the main reasons for the promotion of antibiotics by MRs is the incentives they receive according to the volume of sales.


*“Suppose if they (pharmaceutical companies) give target of INR 100,000 to a MR and if that MR has achieved the target, then next month his/her target will be INR 115,000 and accordingly his/her incentives will also keep on increasing and then they (MRs) also work hard accordingly.”*
(W3)

MRs share incentives with wholesalers. One of the respondents (W2) provided an insight into MRs who have promoted aggressively in their territory to sell antibiotics. They go to nearby cities and share their incentives with wholesalers to purchase antibiotics or other medicines at a cheaper rate. Usually, MRs get the support of wholesalers in their territory, who introduce them to the wholesalers/retailers of nearby towns/cities. In the other cities, they give their product at a cheaper rate to fill the gaps of their targets and compensate for it with the incentive received for meeting their target.

#### 3.3.2. MRs’ Influence over Doctors’ Practice

Respondents (W3 and W4) clearly communicated that MRs cannot influence the prescription practice of doctors; they can only change the brand (trade) name which doctors prescribe. According to them, the promotion of the sale of antibiotics is dependent upon the doctor prescribing the drugs. MRs reach out to a network of doctors in their designated territory, whom they contact when they want to endorse their company’s brands, but they do not influence doctors to prescribe more antibiotics. However, the respondent also elucidated that ‘all the fingers are not same’, implying that some doctors can be promoting antibiotic sales under the influence of MRs and their ‘gifts’.


*“Companies are doing it even for conferences where they want to send doctors, doctors demand money accordingly or some incentives or perks like flight tickets and all.”*
(W3)


*“The normal tendency is that no doctor will give prescription or change the trade name what he/she was writing before, out of courtesy and any other reason, so some obligation has to be there, it can be a gift or any other incentives.”*
(W4)

On the contrary, the opinion of respondent W2 was that doctors prescribe newer antibiotics because of marketing by MRs, and subsequently forget about older antibiotics.


*“Doctors become a target of medical representatives which is one of the key factors of direct sale via them. Doctors say you get this product (gift) I will ensure the sale of your product.”*
(W2)

The ex-SDC shared that, according to trade sources, some doctors are offered expensive gifts like refrigerators, air conditioners, or even foreign trips by companies through MRs for achieving targeted sales in a time-bound manner. Such doctors prescribe antibiotics or high-cost products of such companies to get these gifts in exchange.

#### 3.3.3. Supply Chain of Antibiotics among IHCPs

One respondent (W3) mentioned that the MRs of small and medium companies provide antibiotics to IHCPs. The MRs of multinational companies do not visit IHCPs due to their code of conduct and legal bindings. Multinational companies restrict the sale of products to qualified doctors only, while there are no such obligations for small-scale companies. Here the respondent explained the process by which MRs are able to deliver products to IHCPs at cheaper rates. If the cost of a product is INR 100 and it is not getting sold, then MRs will sell it for INR 60 to IHCPs. They are able to cover the loss through the incentives that they receive from their respective pharmaceutical companies for completing their sales target.


*“All these IHCPs in outskirts receive product of small companies like this, through MRs.”*
(W3)

The respondents (W2, W4) mentioned that MRs are visiting IHCPs, as they are constrained by the area in which they can promote their medicines. One of the wholesalers (W4) mentioned that MRs are the main source of availability of medicines/antibiotics to IHCPs.


*“Yes, they are going (to IHCPs), when area of one MR is limited to one kilometre and even if area is more under him still to every doctor almost 100 MRs are reaching out, so basically there are 200 companies’ brands which doctors has to write, which is not possible.”*
(W4)

The ex-SDC mentioned that IHCPs are present in villages, towns, and cities, and they are the first point of contact for healthcare among low-income households. Most of the small- and many medium-scale pharmaceutical companies supply their products directly to IHCPs at very low rates, as there is no intermediate dealer (e.g., a wholesaler). Moreover, such supply of medicines including antibiotics goes without bills/accounts, escaping taxation.

#### 3.3.4. MRs Sway Retail Sale

Respondent W4 highlighted that MRs also influence sale of retail pharmacies. However, according to the respondent, mostly brands of specific companies are promoted by retailers. Respondents W3 and W4 mentioned that because MRs provide some incentives/gifts to retailers to promote the sale of their company’s specific brand, they tend to dissuade retailers for doctors’ prescribed brands. Respondent W4 mentioned that pharmacists try to convince patients that this written trade name is not available, and substitute it with their recommended brand (trade name) which could be generic and manufactured by a good company, and many patients become convinced.

In addition, respondent W4 mentioned that due to ever-increasing competition in the market, MRs have started to promote medicines by means of various gifts to pharmacists.


*“Retailer come into the picture only when medicines are not getting sold, like if cost of a particular medicine is INR20 and it is not being sold. MRs visit retailer and say that you purchase 30 boxes of it and sell it, I will give you incentives for it.”*
(W3)

However, a positive outcome of this competition is the quality of drugs available in Indian markets. A respondent (W4) explained that unlike earlier days, the manufacturers do not mix or alter content of medicines, although the quality of the product can still be compromised if not the content. Previously this occurred because of a shortage of raw materials, but now due to huge competition in the market, pharmaceutical companies assure the quality of their products and there is no shortage of raw materials or active pharmaceutical ingredients.


*“Now in India instead of medicine you can’t pack chalk powder (something else), that time has gone.”*
(W4)

#### 3.3.5. Institution-Specific Distribution by Pharmaceutical Companies

A respondent (W3) highlighted that multinational pharmaceutical companies target large hospitals, providing drugs like amoxicillin-clavulanic acid and pantoprazole at heavy discounted rates. This is diluting the doctor/MR nexus, and profit is directly going to pharmaceutical companies and hospitals. Hospitals are making huge profits in high volume, as they are using these medicines every day.

The respondent further highlighted that even products like meropenem which are expensive at retail rates are sold to institutions at very reasonable rates. The respondent perceives that institution-specific distribution by pharmaceutical companies leads to excessive and indiscriminate prescription by doctors even without necessity. However, this practice is limited to private healthcare facilities.


*“They (institutions) tell which antibiotics to be manufactured then same will be prescribed and sold. You will see a huge stock of antibiotics in hospital out-and in-patient departments and they are giving where it is not even necessary.”*
(W2)

#### 3.3.6. Third-Party Manufacturing of Drugs and Entrepreneurship Strategies

Respondent W4 shared that most start-up companies compete in business. Owners of these companies mostly have work experience in larger pharmaceutical company as MRs or managers who have worked in the field. Most of these new companies work with third-party manufacturers and start with a few brands/molecules (international non-proprietary name/active pharmacological ingredients) with which they have experience from their job. Thus, according to the respondent, it is not difficult to start a pharmaceutical business.


*“Normally a person who has worked as MR or a manager after a period of time when they have capacity or links or a tendency to work as a company owner, they start their own company by getting a third-party manufacturing done of a few brands (branded-generics).”*
(W4)


*“This can be illustrated from the live examples of many top pharmaceutical companies (names deleted) that their owners were marketing persons before entering in to manufacturing of drugs. Therefore, after getting good marketing experience, the marketing man prefer to start his own company initially via third party manufacturing followed by own unit.”*
(ex-SDC)

Respondents W4 and W3 emphasised that in India a range of products can be manufactured by a third party very easily. In India there is a simple system whereby a manufacturer secures the license of a product as a generic drug and later a trade name can be assigned to the drug. For instance, the respondent mentioned that if a company wanted to sell/distribute amoxicillin with a new trade name, there would be no need to obtain a drugs license for it; the drug could be manufactured through a liaison with a licensed manufacturer of amoxicillin. This is called third-party manufacturing; the product manufactured can then be marketed. Thus, in India there is a rapidly growing trend of pharmaceutical companies using third-party manufacturers.

Respondent W3 explained that traders are engaging in direct product marketing and distribution in small companies who are obtaining their products through third-party manufacturers. For instance, the respondent mentioned his own company, in which manufacturing is done by a third-party agency but product marketing is taken care of by them. The respondent highlighted that because they have established themselves among doctors, they do direct marketing. W3 mentioned obtaining a few antibiotics like amoxicillin-clavulanic acid, cefuroxime, azithromycin, and cefpodoxime, manufactured by a third-party licensed manufacturing unit, and distributing these antibiotics in small cities and towns where doctors usually dispense medicines and direct sales are possible.


*“But now we have started doing direct marketing, we do distribution from here (Delhi) and our team there (other States), handle meeting with doctors.”*
(W3)

However, the ex-SDC clarified that if a marketer intends to sell drugs including antibiotics and relies on third-party manufacturers, a written agreement between the parties’ marketer and manufacturer needs to be submitted to the Licensing Authority. The marketer remains liable for quality and other regulatory compliances along with the manufacturer. A manufacturer who wants to manufacture and sell any medicine under any trade name needs to provide an undertaking to the Licensing Authority based on review of Central Drug Standard Control Organization (CDSCO), literature, and reference books indicating that the proposed trade name is unique and will lead to no confusion or deception in the market. It is not legal to manufacture a drug under multiple trade names simply after obtaining its approval under a generic name.

### 3.4. Understanding of AMR

This theme highlights the understanding of wholesalers regarding the term “antimicrobial resistance”/“antibiotic resistance” and the factors that they believed influence the increased AMR in the community.

#### Knowledge about AMR

One of the respondents (W3) was unaware of the term “AMR”, however on probing for knowledge of antibiotic resistance he could answer somewhat. The respondent (W3) mentioned that the continuous intake of antibiotics leads to the development of resistance in the body and thus the antibiotic is less effective. Respondents W1 and W3 had poor knowledge of AMR/antibiotic resistance, barely understood the consequences of developing AMR, and were not worried about inappropriate or overuse of antibiotics. Respondent W1 elucidated that the community perceives that old antibiotics are not effective and that it is necessary to switch to higher-end antibiotics.


*“Everyone knows that resistance is developing if we don’t properly take medicines or take the incomplete course. But nobody is implementing it properly and they are not paying attention to it.”*
(W1)

W4 mentioned that even among pharmacists, antibiotic misuse is common for business interests and they are not aware of extent of AMR. W4 also mentioned that informal dispensers who run retail pharmacy shops and MRs from non-pharmacy backgrounds have negligible knowledge about AMR/antibiotic resistance. However, W4 emphasised that more than poor knowledge it is a lack of concern about AMR in the community.

## 4. Discussion

This study provides new evidence on the awareness and perceptions of drug wholesalers on supply and utilization of drugs including antibiotics in the community. Much of the earlier evidence on the use of antibiotics in India was from healthcare providers (both formal and informal) and last-mile dispensers, which include retailers and users.

### 4.1. Key Findings with Suggestions/Solutions to Improve the Situation to Contain AMR

Discussions with wholesalers in this study revealed four important findings: (i) they had limited knowledge about wholesale licensing and practice regulations as well as a limited understanding of AMR; (ii) the direct supply and sale of antibiotics to IHCPs occurs; (iii) they facilitated MRs and manufacturers in their strategies to promote antibiotic use in the community; and (iv) they blamed other stakeholders for the unlawful sale and overuse of antibiotics.

Wholesalers had an inconsistent understanding of drug regulations related to stocking for the sale, distribution, and supply of medicines. Their information on this issue did not appear to be based on an understanding of the regulations, but rather general perceptions gathered through conversations with peers in the same business. Among the four wholesalers interviewed in this study, only two were well conversant with the drug licensing regulations. One of the major findings is that wholesalers presumed that they had a legal right to manufacture under any trade name after obtaining approval for a generic name from the Licensing Authority. However, this is not in line with the statuary requirement, as clarified by the ex-SDC. Additionally, the wholesalers had a poor understanding of the term “antibiotic resistance”. They believed that resistance develops in the human body after the repeated use or incorrect dosage of antibiotics.

Wholesalers are involved in the unlawful sale and distribution of antibiotics to IHCPs. In LMICs, the private sector has the predominant role in providing healthcare delivery to the population, which includes both formal healthcare providers and IHCPs [22,23,24]. The IHCPs operate outside the formal health system whilst lacking appropriate qualifications for the healthcare services they deliver, and are not registered as healthcare practitioners with any governing body. In India, IHCPs offer more than 70% of primary healthcare services and dispense medicines for their patients [25]. Primary healthcare services in rural as well as remote, slum areas including difficult-to-reach areas are predominantly catered by IHCPs [26]. This study revealed that wholesalers and small-to-medium-scale manufacturers supply medicines including antibiotics to such IHCPs without sale bills, which is an illegal practice. This practice has been continuing for a while, taking advantage of the weak drug regulatory network in the country.

Our study revealed that wholesalers facilitate the MRs of pharmaceutical companies to sell their products, including antibiotics, to their peers and other retailers. It is common knowledge that pharmaceutical companies have targets for their marketing team, and incentives to MRs are linked to sales volume [27]. Wholesalers facilitate the MRs in achieving their sale targets, and in return MRs share their incentives with them. Finally, respondents of our study blamed all the other actors in the supply chain, from manufacturer to consumer, either for inappropriate use or for promoting antibiotics misuse. Wholesalers blamed pharmaceutical manufacturers, as they promote antibiotic sales through their MRs and supply medicines including antibiotics directly to corporate hospitals at a very discounted price, which promotes misuse by physicians, in order to generate revenue. Respondents perceived that doctors overuse antibiotics and few doctors succumb to the incentives of MRs to prescribe unnecessary antibiotics. Wholesalers also blamed IHCPs for inappropriate antibiotic use because of their inadequate medical knowledge, and blamed retail pharmacists for dispensing the medicines OTC and promoting generic versions of antibiotics for which they have huge profit margins [28].

Our study findings provide opportunities to address some of the issues in wholesale practice that promote AMR in India. Inadequate understanding of drug regulations and inappropriate knowledge of AMR is contributing to promote antibiotic misuse. Regular sensitisation and improving awareness among wholesalers by drug regulators on the storage, supply, and distribution of medicines including antibiotics will be necessary in order to improve their knowledge and competency. Antibiotic resistance and good distribution practices (GDPs), as suggested by WHO [29], should also be made part of such sensitisation programs. The wholesaler is one of the important links of the medicine supply chain, and a mandatory minimum qualification of a diploma in pharmacy for wholesale licensing would be more appropriate. Additionally, the quality and frequency of regulatory checks on distribution by wholesalers should be strengthened to ensure statutory compliance, specifically relating to antibiotics misuse.

Wholesalers facilitate antibiotic misuse involving MRs and IHCPs; however, solutions need to be aimed at the root cause. Pharmaceutical companies should decouple incentives from sales targets to ensure more ethical sales practices. This is especially critical for major companies engaged in the manufacture and sale of antibiotic formulations. Removing an emphasis on sales targets is recognised as a mechanism for reducing the impact of unethical marketing, leading to rational prescribing. Recently, 10 pharmaceutical companies decoupled bonuses from sales volume or removed sales agents altogether [30]. This practice needs to be replicated by pharmaceutical companies in India. Previous studies in India report inappropriate antibiotic practice by IHCPs [26], indicating the need for interventions aimed at them. Integrating IHCPs into the legal healthcare system and having mandatory certified training will be necessary to not only improve access to healthcare but also to optimize antimicrobial prescribing. Recent initiatives from the Government of India to create a trained workforce could be leveraged to improve the quality of care provided by IHCPs [31]. Finally, improving awareness among the general population about AMR will help in recognizing individual actions that promote AMR.

### 4.2. Strengths and Limitations of the Study

To our knowledge, this is the first study from India describing the role of wholesalers in driving AMR. Though the data were collected from the NCR of Delhi, India, the legal provisions relating to wholesale licensing and practices are similar in all states/UTs. However, enforcement may differ from state to state. Nonetheless, more such studies should be carried out in different states of India. Although we interviewed only four wholesalers, interviews were in-depth and generated a range of responses. These four wholesalers supplied to at least 2700 retail stores, indicating their crucial role in antibiotic sales. Convenience sampling was used to reach out to the wholesalers who had experience in the pharmaceutical trade for more than 20 years. However, not many wholesalers would have agreed to give insights of their trade without having trust in the interviewer and the study. The study did not involve the collection and review of evidentiary documents like the checking of wholesale records. However, one of the authors (GLS) was contacted for clarification of some issues arising from the interviews and provided views and experience as an ex-SDC. This is a strength of this study, as the ex-SDC is a senior retired official from a drug regulatory department who provided views and clarified some aspects that were provided by the wholesalers. The information on regulations cited in the paper is authentic and correct due to this source. We tried to minimize the bias to the results, as only one of the authors interviewed all the wholesalers and a social scientist did the transcription, translation, and initial coding of the data. All the themes and sub-themes were finalized by all the authors. Our study is limited to wholesalers who dealt only with human medicines, so the details for veterinary medicine marketing and practices are not detailed in this paper.

## 5. Conclusions

Wholesalers are important stakeholders in managing the AMR problem. Our study results indicate that wholesalers have limited knowledge on wholesale licensing and practice regulations, and a limited understanding of AMR. The interviews conducted indicate that they facilitate the antibiotic misuse activities of pharmaceutical manufacturers, MRs, drug retailers, and IHCPs. Some of the potential solutions aimed at wholesalers include having a minimum education qualification for licensing and mandatory good distribution practices certification programs. Wholesalers’ indirect actions in promoting antibiotic misuse could be alleviated by decoupling incentives by pharmaceutical companies in the form of sales targets to improve ethical MR sales practices, and by efforts to integrate informal healthcare workers into the legal healthcare system by providing proper training.

## Figures and Tables

**Table 1 antibiotics-11-00095-t001:** Themes and sub-themes derived through systematic thematic analysis of in-depth interviews with wholesalers.

Theme	Sub-Themes
Rules and regulations governing wholesale licensing and practice	i. Procurement of a wholesale license ii. Guidelines for wholesale practice iii. Role of drug inspectors and inspections
Antibiotic use and misuse in the community	i. The trend of antibiotic use in the community ii. OTC sale of antibiotics among pharmacists iii. Indiscriminate use of antibiotics by doctors iv. Misuse of antibiotics by the informal healthcare providers (IHCPs) v. Role of wholesalers in the misuse of antibiotics vi. Consumers’ inappropriate antibiotic consumption practice
Potential role of medical representatives (MRs) and pharmaceutical companies in antibiotics misuse	i. MRs’ target-driven approach to marketing ii. MRs’ influence over doctors’ practice iii. Supply chain of antibiotics among IHCPs iv. MRs sway retail sale v. Institution-specific distribution by pharmaceutical companies vi. Third-party manufacturing of drugs and entrepreneurship strategies
Understanding of AMR	vii. Knowledge about AMR

## Data Availability

Aggregate de-identified respondent data will be available after the publication of this study. Requests should be sent to the corresponding author, who will discuss the request with the team and decide whether the data should be shared based on the feasibility, novelty, and scientific rigor of the proposal. All applicants will be required to sign a data access agreement.

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
