# Peer review of "Marketing and Distribution System Foster Misuse of Antibiotics in the Community: Insights from Drugs Wholesalers in India"

_antibiotics, 2022, doi:10.3390/antibiotics11010095_

Round 1

Reviewer 1 Report

This was an interesting manuscript to me.  

The introduction makes the point that India leads the world in Antibiotic consumption - nearly doubling the Defined Daily Doses from the year 2000 to the year 2015.  The study design is simple and straightforward - Structured interviews from 4 wholesalers who supply, in aggregate, approximately 2700 retail stores.  The methods appropriately mention that 3 additional wholesalers declined to participate.   (The use of the word "pretext" in line 109 is perhaps more than you want to say. )

The abstract aptly and succinctly summarizes the content of the paper.  

The Themes and sub-themes appear to be a reasonable breakdown of the topics of the conversations, although it could be revised to better communicate the content of the conversations.  

As for criticisms, I believe the manuscript is about twice as long as it should be.  The abstract nicely summarizes the findings, but the rest of the paper expands perhaps with too much detail on these points.  The introduction does not need to rehash the problem of AMR to the readers of this journal, but it would be  wise to highlight the prominence of India and the role wholesalers contribute to this problem.  Likewise, the methodology and results sections can be condensed to focus the reader’s attention on the key findings. 

I’d suggest that a table similar to Table 1 might present the results in a succinct manner.  A strong table with the key messages could greatly reduce the word count of the results section.  In the discussion section, the manuscript could highlight how these key insights impact antimicrobial resistance and suggest strategies for improvement.

There are some minor-moderate comments on language.  One example:  the term “quacks”  (ie line 314) – is strongly pejorative and should be used cautiously.   Since no clinician considers himself to be a “quack,” recommendations aimed at restricting distribution of antibiotics to quacks won’t have a large impact on antibiotic use. 

Author Response

Point-by-point reply to Reviewer 1 comments

This was an interesting manuscript to me. 

Response: Thank you

The introduction makes the point that India leads the world in Antibiotic consumption - nearly doubling the Defined Daily Doses from the year 2000 to the year 2015.  The study design is simple and straightforward - Structured interviews from 4 wholesalers who supply, in aggregate, approximately 2700 retail stores.  The methods appropriately mention that 3 additional wholesalers declined to participate.   (The use of the word "pretext" in line 109 is perhaps more than you want to say. )

Response: The word ‘pretext’ is changed to ‘indicating’. These 3 wholesalers did not agree to participate, indicating that the wholesalers supply medicines including antibiotics to stakeholders which are suggested by the drug regulators like licensed retail pharmacists and they do not have anything more than that to tell.

The abstract aptly and succinctly summarizes the content of the paper.  

Response: Thank you.

The Themes and sub-themes appear to be a reasonable breakdown of the topics of the conversations, although it could be revised to better communicate the content of the conversations. 

Response: We have not changed the title of themes and sub-themes however; we made several changes in the results section for more clarity and condensed the text as much as possible. Considering the qualitative nature of the study we included detailed results, and this was appreciated by Reviewer 2.

As for criticisms, I believe the manuscript is about twice as long as it should be.  The abstract nicely summarizes the findings, but the rest of the paper expands perhaps with too much detail on these points.  The introduction does not need to rehash the problem of AMR to the readers of this journal, but it would be  wise to highlight the prominence of India and the role wholesalers contribute to this problem.  Likewise, the methodology and results sections can be condensed to focus the reader’s attention on the key findings. 

Response: As suggested, we shortened the introduction.  The qualitative study required a little detail about methodology therefore we have kept the details needed and tried to remove anything which we could. Similarly, for results, the quotes are needed to substantiate the analysis presented for each theme and sub-theme. We condensed the text as much as possible. We did not try to shorten more as reviewer 2 indicated that, “Results section reported all the data described in the methods section and the results were mostly adequately interpreted and given the broader context in the discussion section.”

I’d suggest that a table similar to Table 1 might present the results in a succinct manner.  A strong table with the key messages could greatly reduce the word count of the results section.  In the discussion section, the manuscript could highlight how these key insights impact antimicrobial resistance and suggest strategies for improvement.

Response: Since we have to provide the quotes for each sub-theme and for the main findings, a crisp table indicating all the key messages was difficult to make. We tried to condense the text as much as possible in results section. However, the four important findings have been highlighted in the discussion and each finding is discussed with suggestions/solutions to improve the situation to contain AMR.

There are some minor-moderate comments on language.  One example:  the term “quacks”  (ie line 314) – is strongly pejorative and should be used cautiously.   Since no clinician considers himself to be a “quack,” recommendations aimed at restricting distribution of antibiotics to quacks won’t have a large impact on antibiotic use. 

We have removed the word ‘quack’ and replaced it with informal health care providers (IHCP) and introduced the abbreviation IHCP through the text.

Reply is also uploaded, please see the attachment

Reviewer 2 Report

This interesting study offers a new, fresh perspective on antibiotics’ misuse in low-middle income countries, from a wholesalers’ point-of-view and therefore should be of interest to the readers.

Introduction was mostly well written, covered the main background points and led up to the aim of the study.

Results section reported all the data described in the methods section and the results were mostly adequately interpreted and given the broader context in the discussion section.

However, paper included only a limited number of participants. Furthermore, it only included participants from a single area, which would not be representative of the whole county, especially the one with such diversity like India. This should be mentioned in the limitations section. Could you please also provide an information if there are any differences in legislation and regulation concerning medicines between different Indian states that might be relevant to the wholesalers’ practices.

Paper is also missing the veterinary medicines’ perspective as participants only dealt in human medicines, which should be addressed in limitations section.

Authors used a convenient sample, which should also be addressed in limitations.

Also missing is the influence of the researchers on the data. Authors should give a brief consideration if and how they might have introduced bias to the results, especially since one of the included participants that was interviewed is an author of the paper.

Some minor issues:

  1. Table 1 is missing subheading „iv.“ for the sub-theme „Misusse of antibiotics by the unqualified…“ in the Theme „Antibiotic use and misuse…“
  2. Page 5 line 202 „issues“ is written instead of „issued“.

Author Response

Point-by-point reply to Reviewer 2 comments

This interesting study offers a new, fresh perspective on antibiotics’ misuse in low-middle income countries, from a wholesalers’ point-of-view and therefore should be of interest to the readers.

Response: Thank you

Introduction was mostly well written, covered the main background points and led up to the aim of the study.

Results section reported all the data described in the methods section and the results were mostly adequately interpreted and given the broader context in the discussion section.

Response: We have not changed much in the all the sections of the paper as the reviewer is happy with the presentation. However, we condensed the text as much as possible based on the comments of Reviewer 1.

However, paper included only a limited number of participants. Furthermore, it only included participants from a single area, which would not be representative of the whole county, especially the one with such diversity like India. This should be mentioned in the limitations section. Could you please also provide an information if there are any differences in legislation and regulation concerning medicines between different Indian states that might be relevant to the wholesalers’ practices.

Response: In the limitation section, we have mentioned the point suggested about limited number of participants from a single area. We have added the information in the introduction regarding legislation and regulation for wholesaler in different states is the same as mentioned in national regulation. This point is also included in the limitation section.

Paper is also missing the veterinary medicines’ perspective as participants only dealt in human medicines, which should be addressed in limitations section.

Response: We have included this in limitation section as suggested.

Authors used a convenient sample, which should also be addressed in limitations.

Response: We have addressed this under limitation section as suggested.

Also missing is the influence of the researchers on the data. Authors should give a brief consideration if and how they might have introduced bias to the results, especially since one of the included participants that was interviewed is an author of the paper.

Response: This point is addressed. The findings are based on the interview of the wholesalers. The ex-drug regulator who is also an author was not interviewed but he provided the facts about regulation and was quoted and his comments were for clarifying some issues arising out of the interviews of wholesalers.

Some minor issues:

  1. Table 1 is missing subheading „iv.“ for the sub-theme „Misusse of antibiotics by the unqualified…“ in the Theme „Antibiotic use and misuse…“

Response: In the word document submitted it was iv. Hope this time iv. will be well aligned in the pdf format.

  1. Page 5 line 202 „issues“ is written instead of „issued“.

Response: It is changed as suggested.

The reply is also uploaded

Round 2

Reviewer 2 Report

I would like to thank and congratulate the authors on the improvements of the manuscripts. No further changes are neccesary as the manuscript has been sufficiently improved to warrant publication in Antibiotics journal.

Kind regards